# An Intelligent Networked Car-Hailing System Based on the Multi Sensor Fusion and UWB Positioning Technology under Complex Scenes Condition

**Zhi Wang, Liguo Zang \*, Yiming Tang, Yehui Shen and Zhenxuan Wu**

School of Automobile and Rail Transportation, Nanjing Institute of Technology, Nanjing 211167, China; wangzhi20210729@163.com (Z.W.); yimtang@foxmail.com (Y.T.); x00203180922@njit.edu.cn (Y.S.); a87932885@163.com (Z.W.)

\* Correspondence: zangliguo@njit.edu.cn

**Abstract:** In order to solve the problems of difficulty and long times to pick up cars in complex traffic scenes, this paper proposes an intelligent networked car-hailing system in complex scenes based on multi sensor fusion and Ultra-Wide-Band (UWB) technology. UWB positioning technology is adopted in the system, and the positioning data is optimized by the untraceable Kalman filter algorithm. Based on the environment perception technology of multi sensor fusion, such as machine vision and laser radar technology, an anti-collision warning algorithm was proposed in the process of car-hailing, which improved the safety factor of car-hailing. When the owner enters the parking lot, the intelligent vehicle can automatically locate the owner's position and drive to the owner without human intervention, which provides a new idea for the development of intelligent networked vehicles and effectively improves the navigation accuracy and intelligence of intelligent vehicles.

**Keywords:** intelligent taxi; UWB; lane identification; collision warning

## 1. Introduction

At present, the time consumption caused by parking difficulty is high in China because of increasing rates of car ownership [1]. The main reason is that the driver cannot find parking spaces and their own car. One of the ways to solve the above problems is by an using intelligent vehicle system to make parking efficient [2–4]. In order to improve the efficiency of car pick-up, an intelligent car-hailing system is proposed. After the owner enters the parking lot, the vehicle can automatically locate the owner's position and drive to the owner's side without human intervention. It can solve the problem that the car cannot be found fundamentally and save the time of pick-up.

The intelligent car-hailing system is a comprehensive system with the functions of environment perception, path planning, navigation and decision control [5]. As the basis and premise, environment perception technology is a crucial link to realize intelligent car-hailing, and it is also the basic guarantee of its safety and intelligence [6]. In the environment of dim light, uneven distribution and reflected light intensity, a single sensor for target recognition has inherent defects in sensing range and recognition accuracy [7]. Compared with the traditional single perception, the multi sensor fusion environment perception technology has a higher ability to obtain road environment information that is more accurate and more compatible [8,9]. UWB positioning technology has the advantages of accurate positioning and stable operation, which lay the foundation for the path planning of intelligent car-hailing. The positioning system can effectively locate the dynamic and static targets with high precision and provide a feasible solution for intelligent car-hailing positioning [10,11].

The intelligent car-hailing system can cooperate by providing the whole technology development scheme of the intelligent networked car-hailing system for automobile manufacturers and auto parts enterprises and by realizing the intelligent car-hailing on the

basis of intelligent parking, so as to realize the complete intellectualization of "parking looking for car" and to solve the problems of the difficulty and low efficiency of car picking. It provides a new breakthrough to improve the car pick-up efficiency of large domestic parking lots at the present stage. At the same time, the system can also be further applied to smart cargo trucks in docks, mines and other scenarios, sharing smart cargo platforms, etc., with good market prospects and social value [12,13].

## 2. Development Status at Home and Abroad

Research status abroad: in September 2019, Tesla launched a new function called "smart summon", but this function has a distance limit of 60.96 m and a positioning error of 1 m, which makes the vehicle unable to accurately locate the owner's position, and there are some security risks. According to NHTSA (National Highway Traffic Safety Administration), there have been at least three safety accidents related to Tesla's "smart call" function in the United States, and the short positioning distance is embarrassing in the specific environment of large domestic parking lots.

Domestic research started late, but it has developed rapidly in recent years. In July 2019, the valet parking of Baidu Apollo was able to realize the functions of remote call, automatic queuing, automatic parking and automatic parking. Baidu can realize the perception and control of vehicle environment and trajectory through vehicle measurement sensors. In a certain scene, the relative positioning accuracy is 0.1–0.2 m, the redundancy/omission rate is 0.01% and the L4 is unmanned. However, it has a strong dependence on high-precision cards and high requirements for the application environment. Whether the AVP automatic parking function is still efficient or not, in the case of complex scenes or a lack of on-site arrangement, it is difficult to promote it to a large extent if it cannot solve the problem of high on-site cost.

In October 2020, the function of the NIO OS 2.7 version of the Wei Lai automobile was exposed. Compared with the previous version, we can see from the details that this version has added the function of automatic vehicle call. The new automatic vehicle call function allows users to control the vehicle to move forward and backward for a certain distance near the outside of the vehicle through the app, so as to facilitate users to get on and get off or park the vehicle. Generally speaking, it is more similar to the remote parking function.

In 2021, Liu of Fujian University of technology designed and implemented an intelligent obstacle avoidance path planning system (Figure 1), including the construction of a hardware platform and the implementation of a software obstacle avoidance algorithm. The simulation results verify the effectiveness and feasibility of the combined algorithm. The actual experiment shows that the car can successfully avoid obstacles from the planned initial position and reach the designated position on the planned path.

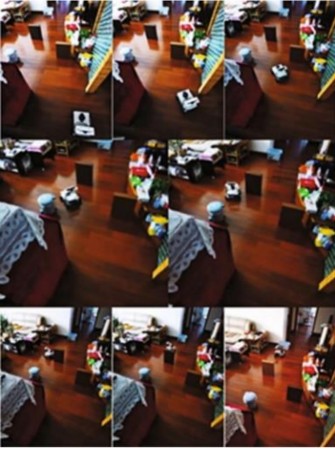

**Figure 1.** Path map of intelligent car-hailing.

## 3. Intelligent Online Car-Hailing System in Complex Scenes

The framework of the intelligent networked car-hailing system is shown in Figure 2. This paper will describe it from three aspects: navigation and positioning, environment perception and path planning.

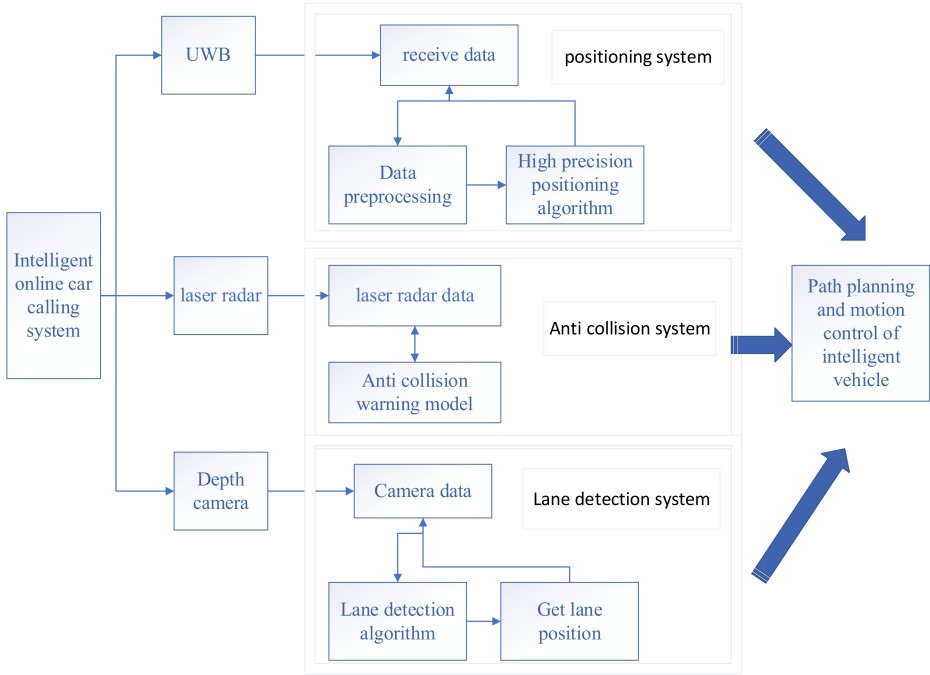

**Figure 2.** Frame diagram of the car-hailing system.

### 3.1. Navigation and Positioning

Positioning generally has the following methods: GPS, Ultrasonic, WIFI, UWB, RFID, FMCW and so on. With the development and modernization of the global navigation satellite system, it has made a great breakthrough in positioning and navigation and has been widely used [14,15]. However, GPS is susceptible to weather, earth rotation, satellite movements, cloud flow and other factors [16]. Additionally, the indoor signal strength is weak and the accuracy cannot meet the demand. The ultrasonic wave easily produces a multipath effect and NLOS error, and the frequency is affected by the Doppler effect and temperature [17]. WIFI is a wireless local area network composed of wireless access points, but its positioning error is large and susceptible to interference [18]. RFID can achieve centimeter-level positioning within a few milliseconds, but it operates at a short range and has no communication capability [19]. The FMCW radar is limited by transmit power, antenna efficiency and receiver sensitivity. The FMCW signal bandwidth also limits the maximum range of the radar, leading to concessions in range resolution. The pairs of different indoor positioning methods are shown in Table 1.

UWB positioning technology has the advantages of a short working time, strong penetration, high time resolution and high matching accuracy. It is initially decided to adopt UWB positioning technology, and the trilateration trilateral measurement method will be used to locate the vehicle owner's position. In order to improve the positioning accuracy of UWB and ensure the stability of data, the unscented Kalman filter algorithm is used.

**Table 1.** Common indoor positioning methods.

| Positioning Mode | Comparison of Common Indoor Positioning Methods | | | | |
|---|---|---|---|---|---|
| | Accuracy | Penetrability | Anti-Interference | Layout Complexity | Cost |
| ZigBee location | ★★☆☆☆ | ★★★★☆ | ★★★☆☆ | ★★☆☆☆ | ★★★☆☆ |
| RFID location | ★★★★☆ | ★★★☆☆ | ★★☆☆☆ | ★★☆☆☆ | ★★☆☆☆ |
| WI-FI location | ★☆☆☆☆ | ★★★☆☆ | ★★★☆☆ | ★☆☆☆☆ | ★☆☆☆☆ |
| UWB location | ★★★★★ | ★★★★☆ | ★★★★☆ | ★☆☆☆☆ | ★★☆☆☆ |

### 3.2. Environmental Perception

Single sensor for target recognition has inherent defects in sensing range and recognition accuracy. For example, the light transmission efficiency of the atmosphere has a great impact on the LiDAR, which cannot work all day and cannot work normally in bad weather. The directivity of ultrasonic ranging is poor, and the detection accuracy is low in bad weather. Infrared ranging has a low resolution, small detection range and short operating range [20,21]. Therefore, it is difficult to provide a comprehensive description of the road environment only by a single sensor.

In this paper, the fusion technology of machine vision and LiDAR multi-sensor is adopted to improve the accuracy of the sensing information. In terms of environmental perception, LiDAR is mainly used for obstacle detection, avoidance, mapping and 3D motion capture. The lane line is detected by machine vision technology to ensure the safe driving of vehicles. Firstly, the image of the lane is corrected and preprocessed, and then gray and Gaussian fuzzy are processed in turn. Then, the interested area is selected according to the lane line detection in the previous frame. Finally, the edge detection is carried out based on the improved sparrow search algorithm, and the lane line is extracted by Hough transform.

### 3.3. Path Planning

A* algorithm is a typical global path planning algorithm, and its basic implementation process is as follows: calculate the f value of each sub node from the starting point, select the sub node with the smallest f value as the next search point, and iterate repeatedly until the next sub node becomes the target point. Based on A* algorithm, Anthony Stentz proposed D* algorithm in 1994. D* algorithm is a reverse incremental search algorithm; that is, the reverse algorithm searches from the target point to the starting point step by step. Incremental search—that is, the algorithm—calculates the distance information H(x) of each node in the search process. In a dynamic environment, if there are obstacles that cannot be searched on the path in advance, the algorithm will redesign the path of the current state point according to the distance information of each point obtained in advance and carry out appropriate path planning [22–24]. Finally, according to the mastery degree of environmental information, it makes the corresponding judgment.

## 4. Key Technologies of Intelligent Networked Car-Hailing System

As shown in Figure 3, three key technologies must be solved for the normal operation of the system: (1) high precision positioning technology; (2) lane detection technology; and (3) the model and algorithm of vehicle anti-collision warning.

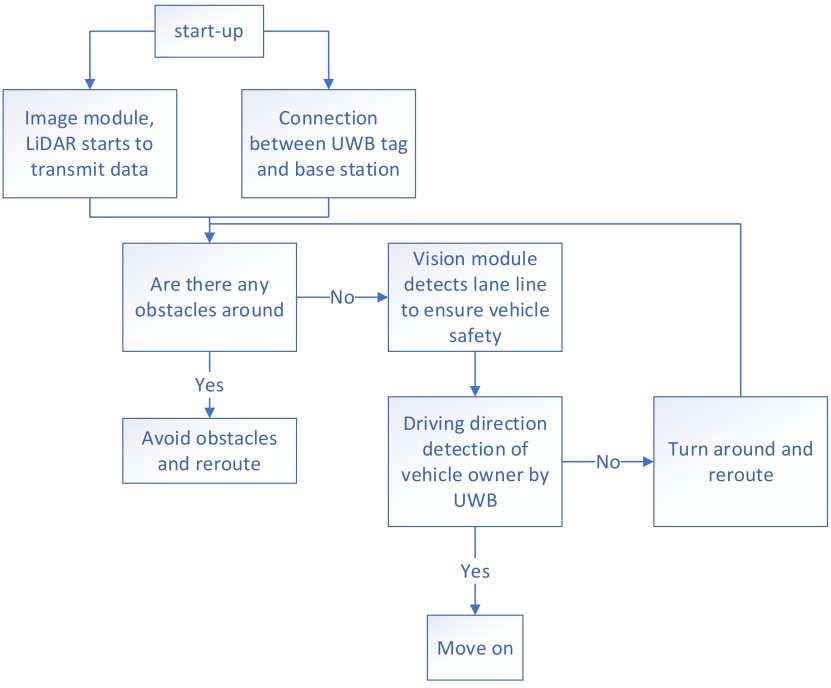

**Figure 3.** Car-hailing flow chart.

*4.1. High Precision Positioning Technology*

The traditional three-sided measurement method may cause the three locating circles to be unable to be intersected at one point due to the different power consumption and the existence of the multipath effect of the base station, which will reduce the positioning accuracy. Therefore, the standard trilateral measurement method is improved.

In most cases, the trilateral method will intersect in an area, as shown in Figure 4. Set the intersection of three circles as $(x_a, y_a)$,$(x_b, y_b)$,$(x_c, y_c)$ and select the center of gravity $\left(\frac{x_a+x_b+x_c}{3}, \frac{y_a+y_b+y_c}{3}\right)$ of the triangle surrounded by three points to represent the position of the label. Obviously, the size of the area surrounded by three circles has a great influence on the location accuracy of the label, and experiments show that the smaller the area surrounded, the higher the accuracy. Therefore, the area can be used as a weight to measure the credibility of the positioning point.

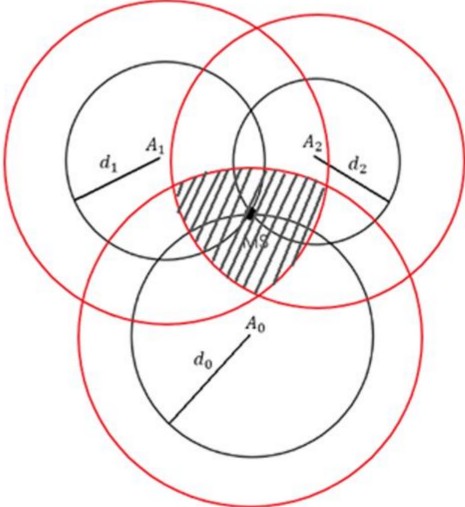

**Figure 4.** Improved trilateration method.

There is a positive correlation between the intersection area of three circles and the triangle area surrounded by the three intersection points. The relationship between the intersection area of three circles and the accuracy of location can be approximated by the relationship between the triangle area surrounded by the three intersection points and the accuracy of location. The smaller the area of the triangle is, the more accurate the anchor point is and the larger the weight is. The three sides of the triangle formed by the intersection point, respectively, are:

$$
\begin{cases}
l_0 : y - y_a = \frac{y_a - y_b}{x_a - x_b}(x - x_a) \\
l_1 : y - y_b = \frac{y_b - y_c}{x_b - x_c}(x - x_b) \\
l_2 : y - y_c = \frac{y_a - y_c}{x_a - x_c}(x - x_c)
\end{cases}
\tag{1}
$$

Triangle area:

$$
S = \int_{x_c}^{x_b} (l_1 - l_2)dx + \int_{x_b}^{x_a} (l_0 - l_2)dx
\tag{2}
$$

Suppose that UWB obtains n groups of valid data every 0.1 s and the triangle area enclosed by the intersection under each group of data is s. The following weight function can be defined to represent the credibility of the location:

$$
x_{c,j} = \frac{x_i}{S_i}
\tag{3}
$$

$$
y_{c,j} = \frac{y_i}{S_i}
\tag{4}
$$

The optimal position is obtained by standardization:

$$
x = \frac{\sum_{i=1}^{n} \frac{1}{S_i} x_i}{\sum_{i=1}^{n} \frac{1}{S_i}}
\tag{5}
$$

$$
y = \frac{\sum_{i=1}^{n} \frac{1}{S_i} y_i}{\sum_{i=1}^{n} \frac{1}{S_i}}
\tag{6}
$$

Above all, the unknown point position is estimated for many times, the location point is determined by using the weight of each position and the optimal location point is obtained by the unscented Kalman filter algorithm with an adaptive factor. It not only solves the problem that the standard trilateral measurement method cannot meet at one point, but it also overcomes the influence of the multipath effect and NLOS error on positioning to a certain extent, increases the reliability of positioning and realizes the high-precision positioning of the human vehicle.

In order to verify the rationality of the positioning and algorithm, static tests and dynamic tests were carried out, respectively. The test was carried out in a 10 m × 10 m LOS environment using the DWM 1000 radio frequency module produced by Decawave. The typical bandwidth is 500 MHz, the transmission power adjustment range is −62−−35 dBm/MHz and a communication rate of 110 kB/s is selected.

In the static positioning test, at a positioning speed of 10 times per second, the lateral and longitudinal errors are within 10 cm, the highest positioning accuracy is within 7 cm, the RMSE (root mean square error) is reduced by 12.74% and the MaE (average absolute Error) is reduced by 21.58%, as shown in Table 2.

**Table 2.** Static positioning test results.

| Direction | Value | Traditional Algorithm | Improve Algorithm |
|---|---|---|---|
| | Positioning accuracy/cm | 7.09 | 5.57 |
| X direction | Root mean square error/cm | 5.18 | 4.52 |
| | Mean absolute error/cm | 7.09 | 5.56 |
| | Positioning accuracy/cm | 8.94 | 6.82 |
| Y direction | Root mean square error/cm | 6.81 | 5.73 |
| | Mean absolute error/cm | 8.95 | 6.82 |

The comparison between the static positioning result and the traditional positioning algorithm is shown in Figure 5. It can be seen from the experimental results that this work proposes an improved trilateral measurement method. The absolute positioning accuracy and relative positioning accuracy have been significantly improved.

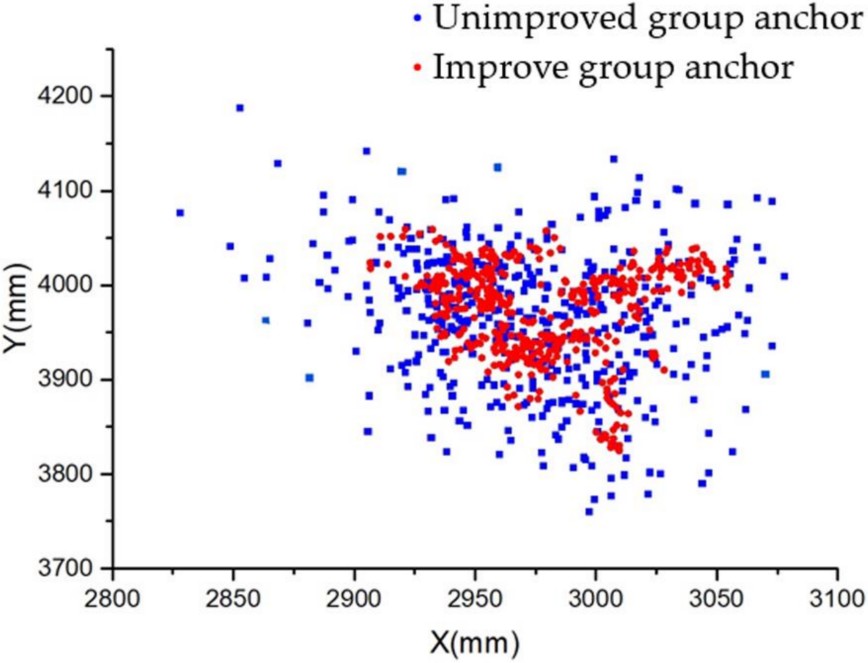

**Figure 5.** Comparison of static positioning.

In the dynamic positioning S-shaped test, the maximum deviation in the horizontal and vertical directions is within 20 cm and the error of most points is within 10 cm. Compared with the ideal point (3000, 4000), the average error in the x-direction in the unimproved group is 7.091 cm and the maximum error is 17.208 cm. The average error in the y direction is 8.947 cm and the maximum error is 23.987 cm. In the improved group, the average error in the x direction is 5.566 cm and the maximum error is 10.225 cm. The average error in the y direction is 6.817 cm and the maximum error is 18.038 cm. The dynamic positioning trajectory is shown in Figure 6.

In dynamic positioning, the maximum deviation in the x direction is 12.4 cm and the maximum deviation in the y direction is 15.0 cm. The error of most points is within 10.0 cm, and the trajectory fluctuates little, which is basically consistent with the ideal trajectory.

In order to verify the optimization effect of the algorithm of this work on UWB positioning data and the ability to correct NLOS errors, a simulated NLOS test was carried out in a laboratory environment of 10 m × 10 m. A UWB device with a transmission power of –45 dBm/MHz is adopted, with a communication rate of 6.8 m/s, a center frequency of 6489.6 MHz, a frequency band of 6240–6739.2 MHz and a bandwidth of 499.2 MHz. It complies with IEEE 802.15.4-2011 protocol. The test scenario is shown in Figure 7. The base

station is placed along the vertices of an equilateral triangle with a side length of 6 m, and the dashed line represents the movement trajectory of the tag MS. A metal block is inserted between the base station and the track. When the tag, the metal block and the base station are on the same straight line, the metal block with a dielectric coefficient greater than 1 will interfere with the propagation of the direct single-path component DP, forming a strong NLOS environment. The traditional positioning algorithm and the algorithm of this work were used to carry out the positioning test, and the test results are shown in Figure 8.

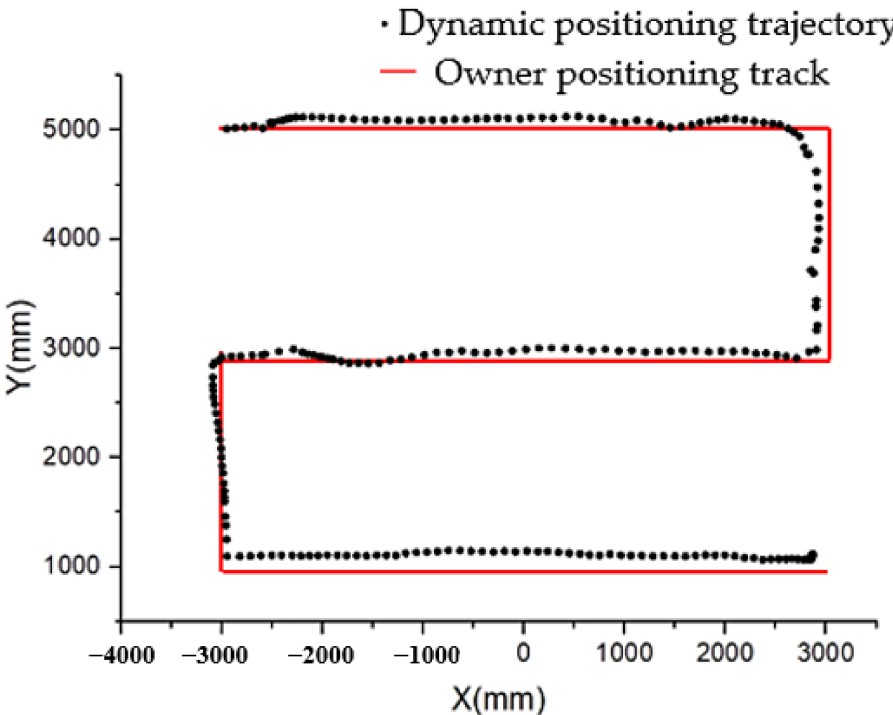

**Figure 6.** Dynamic positioning trajectory.

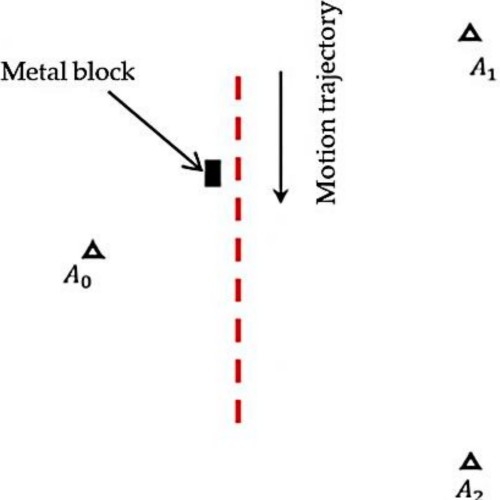

**Figure 7.** Test environment.

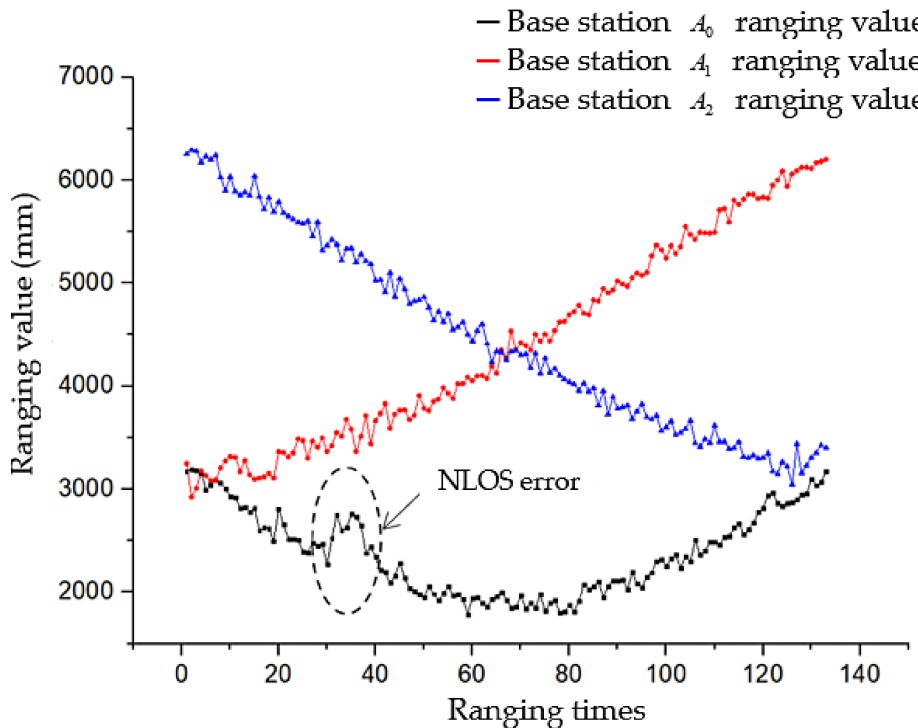

**Figure 8.** Base station ranging value.

It can be seen from Figure 4 that in the 30th to 40th ranging, the base station was affected by the metal block and the ranging value suddenly changed, which is shown in the positioning result, as shown in Figure 9.

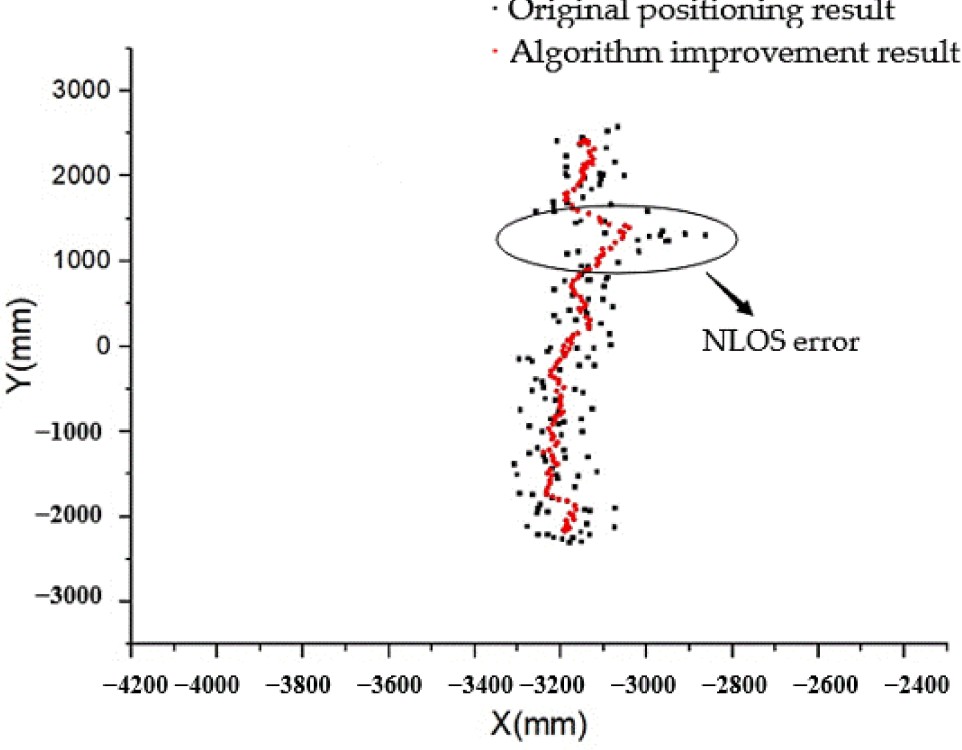

**Figure 9.** Positioning results of traditional algorithms.

The black dots in Figure 5 are the positioning results of the traditional algorithm. It can be seen that the data began to be abnormal at $y = 150$ cm, and then there was a large error in the positioning and the maximum error value reached 33.589 cm. The red dot in the figure is the positioning of the work algorithm. As a result, it is obvious that the data fluctuation is smaller and the data mutation at the original NLOS error is relatively smooth. The maximum error is only 15.883 cm, which is 52.7% lower than the maximum error of the traditional algorithm. The RMSE value and MAE value are shown in Table 3.

**Table 3.** Test results.

| Direction | Value | Unimproved Group | Improvement Group |
|---|---|---|---|
| X direction/cm | RMSE | 7.98 | 5.51 |
| | MaE | 6.18 | 4.25 |

Experiments show that the algorithm proposed in this work improves the positioning accuracy of UWB, solves the problem of low positioning accuracy in complex scenes and realizes the real-time accurate positioning of people and vehicles.

*4.2. Lane Detection Technology*

The premise of lane detection is to detect the edge of the lane. Because of the uneven distribution of light in the parking lot and the strong reflection of the ground, the commonly used edge detection methods easily judge the noise as the edge, so the threshold needs to be manually selected. This paper proposes a lane detection method based on an improved sparrow search algorithm, which reduces the interference of ambient light by dynamically selecting the region of interest (ROI). The optimization of parameters such as step size and warning threshold can avoid noise, quickly and accurately find the edge area of lane line, improve the accuracy of lane line detection and ensure the safety and standardization of calling vehicles.

The main flow chart (Figure 10) and steps of edge detection of the improved sparrow search algorithm are as follows:

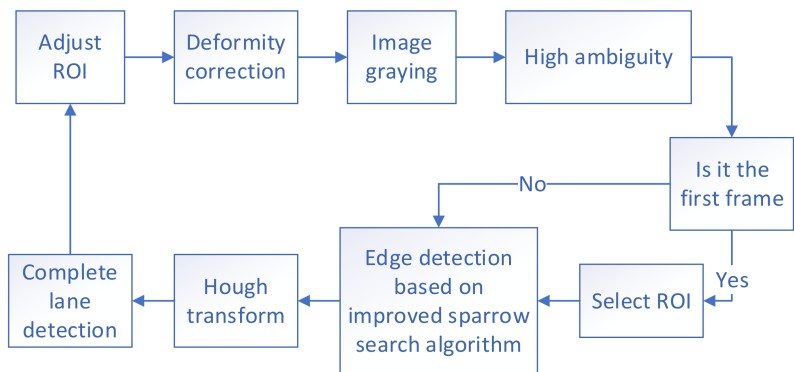

**Figure 10.** Flow chart of lane detection algorithm.

(1) Input the pre-processed image, and each pixel is regarded as a location where the sparrow can forage. Initialize the sparrow position: randomly distribute n sparrows to the pixels of the image.

(2) A fixed proportion of sparrows were selected as the discoverers to provide direction for the whole sparrow species. The location updating formula is as follows:

$$m_{i,z}^{t+1} = \begin{cases} m_{i,z}^t * exp(\frac{-i}{\alpha * n_{max}}), C_{(x,y)} < ST \\ m_{i,z}^t + Q, C_{(x,y)} \geq ST \end{cases} \tag{7}$$

where $m_{i,z}^t$ is the position of the $i$ sparrow in generation $t$ in the $z$ dimension of the image, $\alpha$ is the random number in the value space 0.1, $n_{max}$ is the maximum number of iterations, $Q$ is the random number in the standard normal distribution, $ST$ is the alert threshold and $C_{(x,y)}$ is the fitness of the position, which is defined as follows:

$$C_{(x,y)} = \frac{1}{255} \times max \begin{bmatrix} H(x-1,y-1) - H(x+1,y+1) \\ H(x-1,y+1) - H(x+1,y-1) \\ H(x,y-1) - H(x,y+1) \\ H(x-1,y) - H(x+1,y) \end{bmatrix} \tag{8}$$

In the formula, $H(x,y)$ represents the gray value of the pixel at $(x,y)$ position. The gray value of pixels in the edge region changes greatly and has better adaptability. When $C_{(x,y)} < ST$, it doesn't reach the warning threshold and it carries out an extensive search. When $C_{(x,y)} \geq cST$, it reaches the warning threshold and randomly moves in a small range according to the standard normal distribution.

(3) In each generation of sparrows, except the sparrow selected as the discoverer, it is regarded as the follower. The follower follows the discoverer to forage, and its location updating formula is as follows:

$$m_{i,z}^{t+1} = \begin{cases} Q * exp\left(mw_{i,z}^t - m_{i,z}^t\right), i > \frac{n}{2} \\ mb_{i,z}^t - \frac{1}{D}\sum_{d=1}^{D}\left(rand\{-1,1\} * \left|mb_{i,z}^t - m_{i,z}^t\right|\right), i \leq \frac{n}{2} \end{cases} \tag{9}$$

$mw_{i,z}^t$ is the lowest global fitness position of the current sparrow, $mb_{i,z}^t$ is the highest fitness position of the current discoverer and $D$ is determined by the location dimension. Since the input image is a two-dimensional image, the value of $D$ in this work is 2. When $i > \frac{n}{2}$, it indicates that the adaptability of the position of the $i$ follower is relatively low and it needs to fly to other places. When $i \leq \frac{n}{2}$, the follower will fly to the optimal position of the current discoverer at random.

(4) All sparrows, except those selected as discoverers, are regarded as followers who follow discoverers to forage. The formula for updating its position is as follows:

$$m_{i,z}^{t+1} = \begin{cases} mr_{i,z}^t + \beta\left|m_{i,z}^t - mr_{i,z}^t\right|, C_{(x,y)} \neq C_{max} \\ m_{i,z}^t + K\left(\frac{\left|m_{i,z}^t - mw_{i,z}^t\right|}{\left(C_{(x,y)} - C_{min}\right) + \varepsilon}\right), C_{(x,y)} = C_{max} \end{cases} \tag{10}$$

In the formula, $mr_{i,z}^t$ is the highest global fitness position of the current sparrow, the parameters of $\beta$ control step follow the standard normal distribution, $K$ is the random number between $[-1, 1]$, $C_{max}$ is the current global maximum fitness, $C_{min}$ is the current global minimum fitness and $\varepsilon > 0$ in order to prevent the denominator from being 0. When $C_{(x,y)} \neq C_{max}$, the sparrow flies to the best position in the whole world. When $C_{(x,y)} = C_{max}$, the sparrow will fly to a nearby position.

(5) Continue to iterate until the maximum number of iterations is reached. In the edge detection, the output image is a binary image. The pixel value of the position where the sparrow stays is set to 1, and the pixel value of the position where the sparrow does not stay is set to 0. Finally, the image is output, and the effect of the output image is shown in Figure 11.

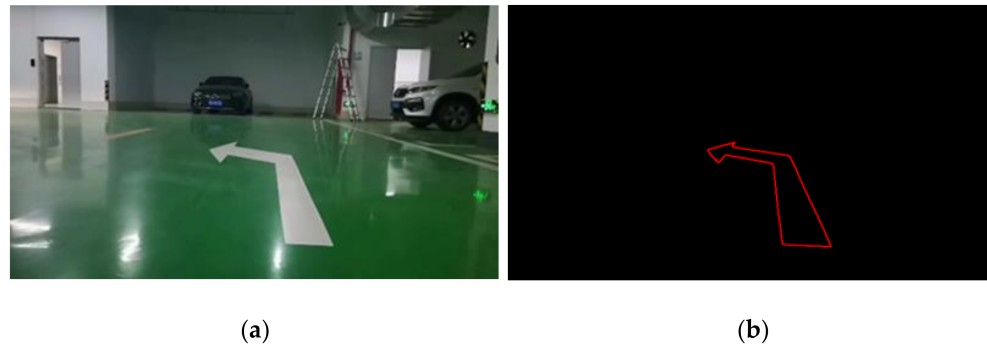

(**a**)                                                         (**b**)

**Figure 11.** Before and after edge detection. (**a**) Image to be detected. (**b**) Edge detection renderings.

The last step of edge detection also detected some interference factors; there is a need to further operate the image after edge detection to find the lane line. In most cases, the lane line is a straight line, so this paper uses Hough transform to extract the lane line.

Input the two-dimensional image. In a Cartesian coordinate system, the function expression of straight line is $y = kx + b$ and the expression converted to the Hough coordinate system is $b = -xk + y$. As shown in Figure 12 below, a point on a straight line in a Cartesian coordinate system is transformed into a Hough space and is intersected at a point.

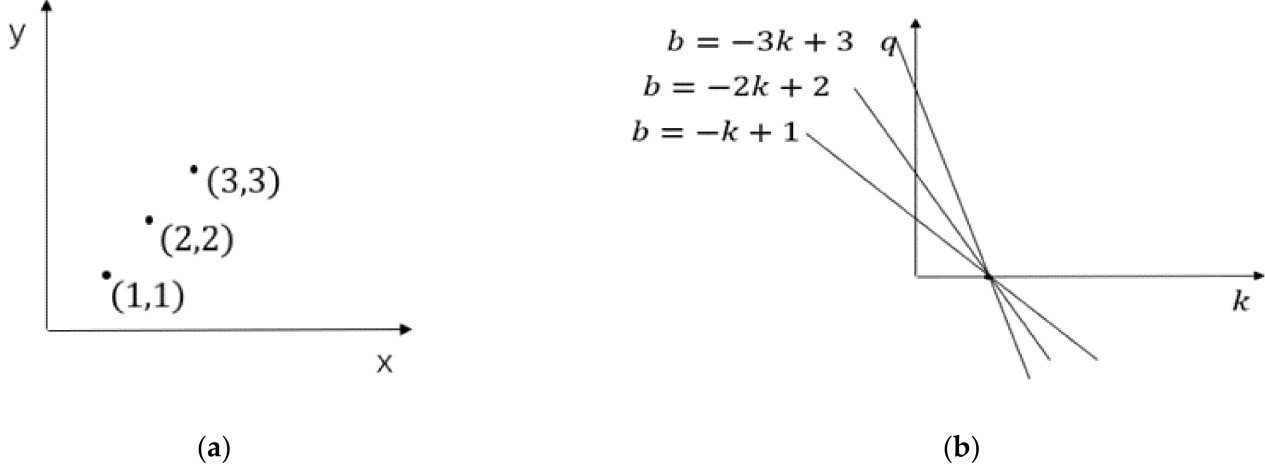

(**a**)                                                         (**b**)

**Figure 12.** Schematic diagram of a Hough transformation in a cartesian coordinate system. (**a**) Cartesian coordinate system. (**b**) Hough Cartesian coordinates.

However, in the case of $k = \infty$, it is not convenient for the straight line to be expressed in the Cartesian coordinate system, so it needs to be converted to the polar coordinate system that $x \cos \theta + y \sin \theta = \rho$, and then the Cartesian coordinate system is converted to the Hough coordinate system and the function expression after conversion is $\rho = \cos \theta \, x + \sin \theta \, y$. As shown in Figure 13 below, a line in the polar coordinate system is transformed upward into the Hough space and intersects at a point:

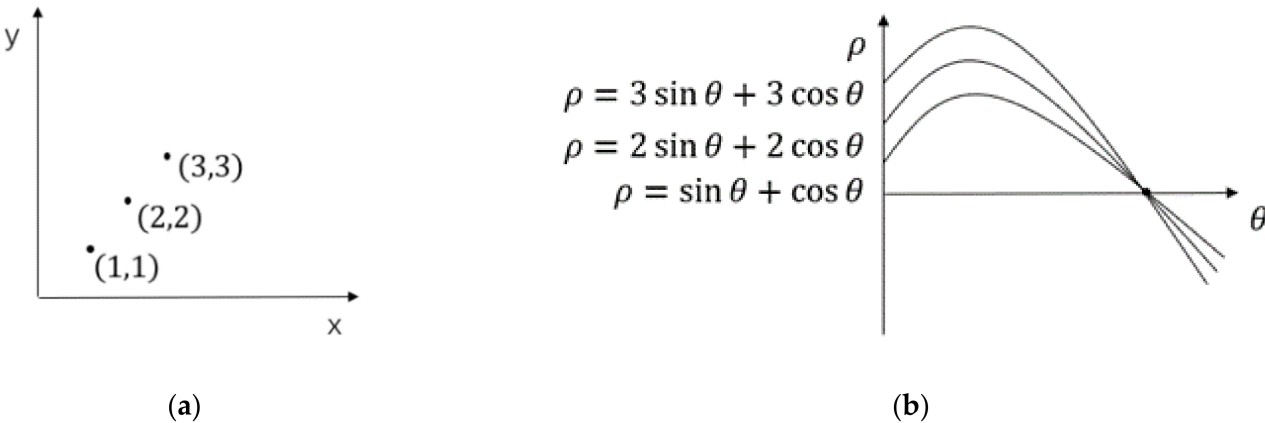

(a)                                        (b)

**Figure 13.** Schematic diagram of a Hough transformation in a polar coordinate system. (**a**) Cartesian coordinate system. (**b**) Hough Cartesian coordinates.

Therefore, lane lines can be well fitted through Hough transform, and the effect of lane line detection can be achieved, as shown in Figure 14.

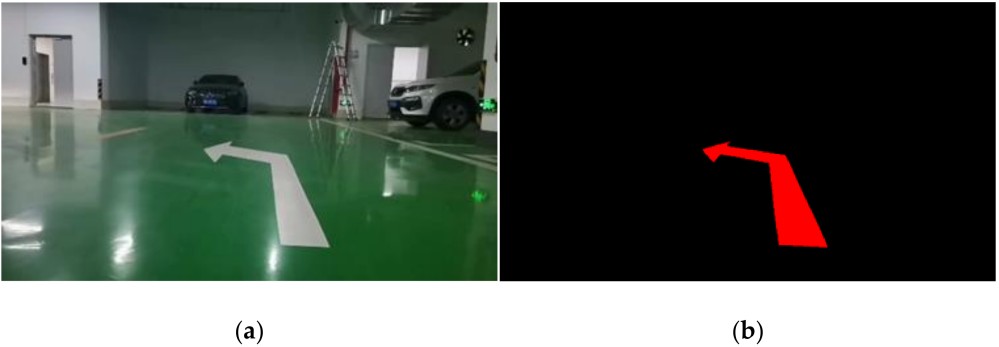

(a)                                        (b)

**Figure 14.** Before and after lane line detection. (**a**) Image to be detected. (**b**) Renderings of lane line detection.

In order to verify the effect of the lane line detection and recognition method based on the improved sparrow search algorithm proposed in this paper, this paper performs lane line detection on the same computer on the videos taken by three groups of actual vehicles driving on the road. The results are shown in Table 4.

**Table 4.** Test results of this method.

| Video Number | Number of Frames | The Number of Frames Correctly Detected | Detection Accuracy |
|:---:|:---:|:---:|:---:|
| 1 | 1260 | 1210 | 96.03% |
| 2 | 1550 | 1482 | 95.61% |
| 3 | 1700 | 1604 | 94.35% |

For three groups of the same video on the same computer, the traditional edge detection method based on the canny operator is used to detect lane lines. The results are shown in Table 5.

**Table 5.** Edge based on canny operator detection method detection result.

| Video Number | Number of Frames | The Number of Frames Correctly Detected | Detection Accuracy |
| --- | --- | --- | --- |
| 1 | 1260 | 1082 | 85.87% |
| 2 | 1550 | 1296 | 84.61% |
| 3 | 1700 | 1382 | 83.29% |

In Tables 4 and 5, video 1 is a video of a vehicle driving on a highway during the day, video 2 is a video of a vehicle driving on a highway on a cloudy day and video 3 is a video of a vehicle driving on a highway under strong light. From the simulation results in Tables 4 and 5, it can be seen that under the same number of frames, the lane line detection and recognition method based on the improved sparrow search algorithm proposed in this paper has a higher detection and recognition rate.

### 4.3. Vehicle Collision Warning Model and Algorithm

The safety of vehicle operation is affected by many factors such as meteorological conditions and road conditions. The traditional safety distance calculation model is only calculated by speed limit, which leads to an inaccurate alarm. In this paper, an improved Berkeley model is proposed; that is, the vehicle forward and backward motion parameters are analyzed using LiDAR, and the effect of braking deceleration on displacement is ignored [25,26].

$$\int_0^{t_3} \left( \int_0^t \frac{a_{max}}{t_s} t \, dt \right) dt \tag{11}$$

(1) When the car in front moves at a constant speed and the car behind decelerates to 0, the required time is as shown in Figure 15 for the velocity displacement variation of the two vehicles.

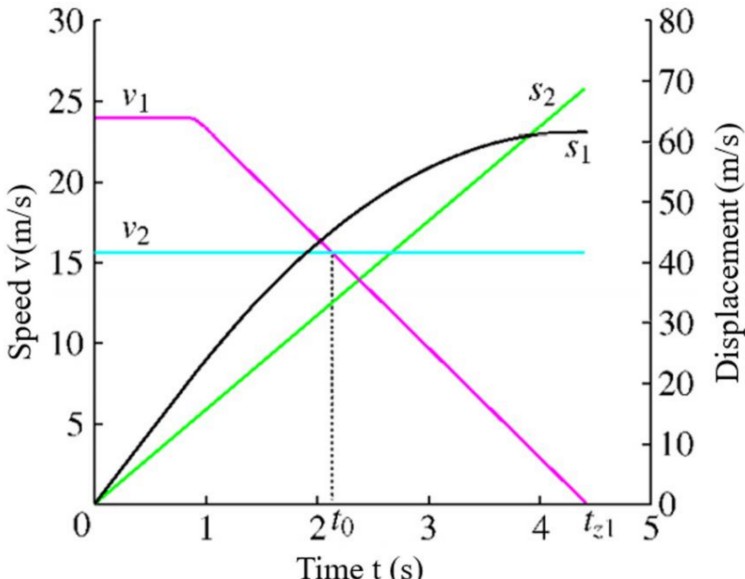

**Figure 15.** Velocity—displacement variation diagram.

When $t_0$, the speed of the vehicle behind, decreases to the same as that of the vehicle in front, the safety distance between the two vehicles is:

$$D_b = s_1(t_0) - s_2(t_0) + d_0 \tag{12}$$

(2) When the deceleration time of the rear vehicle to zero is greater than or equal to the time when the speed deceleration time of the front vehicle reaches zero, the velocity-displacement changes of the two vehicles are shown in Figure 16.

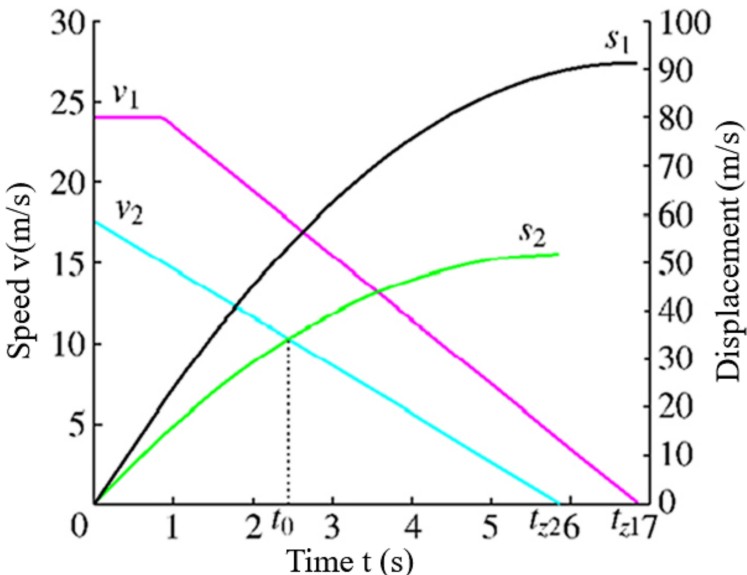

**Figure 16.** Velocity—displacement variation diagram.

When the vehicle has decelerated to a static state at $t_{z2}$ time because of $t_{z1} > t_{z2}$, the following vehicle still keeps a certain deceleration state and moves forward. The safe distance is:

$$D_b = s_1(t_{z1}) - s_2(t_{z2}) + d_0 \tag{13}$$

(3) When the time $t_{z1}$ that the rear vehicle decelerates to zero is smaller than the time $t_{z2}$ that the front vehicle decelerates to zero, the velocity-displacement changes of the two vehicles are shown in Figure 17.

At $t_0$ time, the speed of the two vehicles is the same and the safety warning distance is:

$$D_b = s_1(t_0) - s_2(t_0) + d_0 \tag{14}$$

Synthesizing the above three situations, different calculation formulas of safety warning distance are selected at the right time.

$$D_b = \begin{cases} s_1(t_0) - s_2(t_0) + d_0 & t_{z1} < t_{z2} \\ s_1(t_{z1}) - s_2(t_{z2}) + d_0 & t_{z1} \geq t_{z2} \end{cases} \tag{15}$$

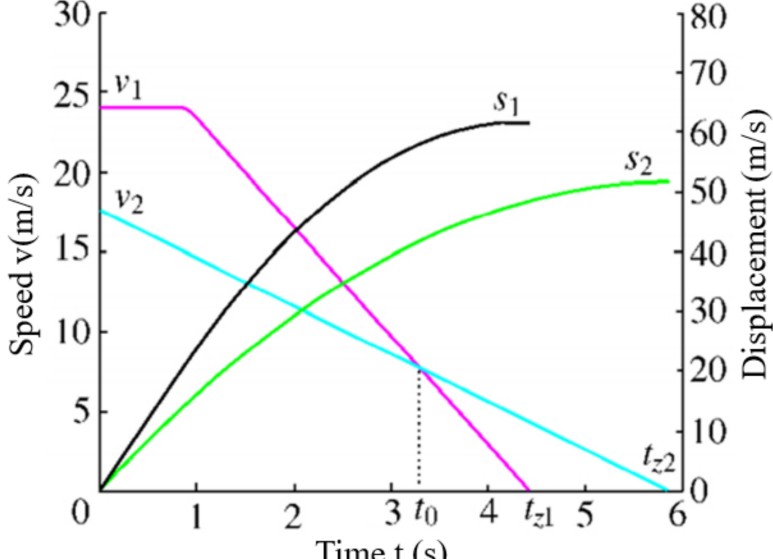

**Figure 17.** Velocity—displacement variation diagram.

In order to remedy the defect of the Berkeley model neglecting side safety, this paper proposes a new side safety distance model for driving on curved roads (Figure 18).

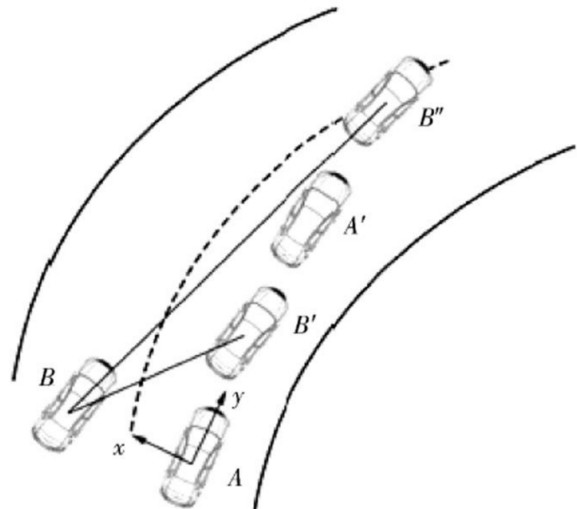

**Figure 18.** New lateral safe distance model.

For the model with the angle of the two cars and the speed of the car as the main parameters, the lateral critical safety distance model is established, according to the angle between the two cars $\theta = arctan\frac{d_2 - d_1}{D}$ (D ipsilateral two laser sensors for vehicle distance, $d_1$, $d_2$; respectively different obstacle distance from the laser sensor), and the relative velocity was carried out on the side of the vehicle driving state, an analysis of lateral safe distance:

$$d_0 = sin\theta\left(v_B t + \frac{1}{2}a_B t^2\right) + 0.94\begin{cases} t = t_1, X_{By} < X_{AA'} \\ t = t_4, X_{By} > X_{AA'} \end{cases}$$

($X_{By}$ is the longitudinal traveling distance of car B and $X_{AA'}$ is the longitudinal traveling distance of car A.) Compared with the traditional model, this model takes into account the influence of other vehicles on the side safety in the process of self-vehicle turning and lane change. It has fast signal transmission, strong anti-interference ability and can predict the lane change of vehicles in different lanes in real time, thus improving the accuracy of early warning.

Simulation analysis of the improved Berkeley model and the traditional Berkeley model based on MATLAB (the left picture in Figure 19 is the traditional Berkeley model, and the right picture is the improved model of the work). The simulation data shows that at different vehicle speeds, the improved Berkeley model has an earlier warning, and this advantage becomes more obvious as the vehicle speed increases. When the relative vehicle speed is higher, the safety warning distance in the improved model is compared with the safety warning in the traditional model. The distance is relatively small, and the difference is about 1 to 2.5 m. Compared with the traditional model, the improved Berkeley model can effectively prevent false alarms and realize that the safety warning distance changes according to the changes of the front and rear vehicle motion parameters.

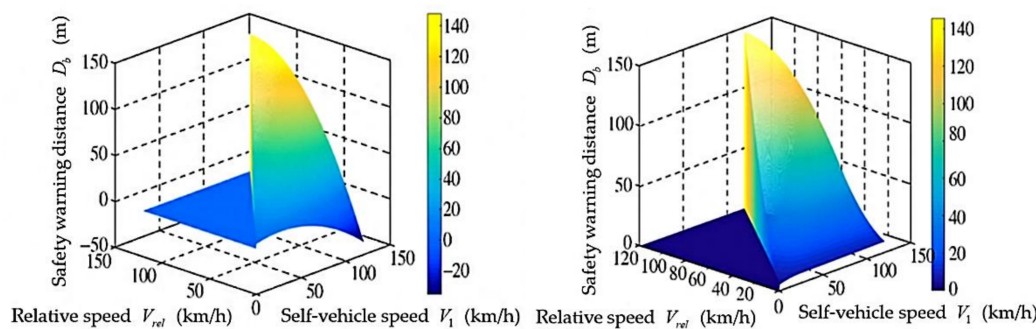

**Figure 19.** Comparison of the early warning distances of the improved Berkeley model.

The lateral safety distance model of the curve is simulated based on MATLAB, as shown in Figure 20.

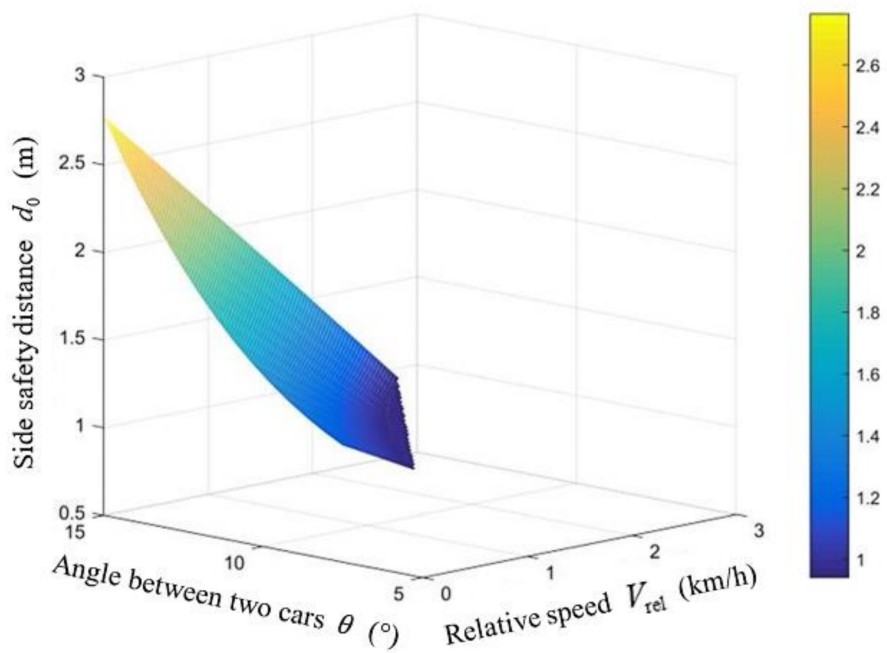

**Figure 20.** Curve lateral safety distance model early warning distance.

When passing railway crossings, sharp bends, narrow roads and narrow bridges, the maximum speed of motor vehicles shall not exceed 30 km per hour, which is about 8.3 m/s. So, take $V_A$ = 25 km/h, which is about 6.94 m/s.

In order to ensure that the two vehicles are in a safe state when they are stopped, the safe distance between the vehicle and the vehicle in front of it after stopping is generally taken as 2 to 5 m. In view of the fact that both vehicles are driving on curves, in order to ensure that the two vehicles still have a safe distance after driving for a period of time, the longitudinal critical safety distance is $S_0$ = 3.5 m.

The results show that when the reference speed is 25 km/h and the longitudinal critical safety distance is 3.5 m, the lateral safety distance and lateral critical safety distance change significantly with the change of the included angle and relative speed of the two vehicles. When the driving speed is low, the change of the lateral critical safety distance is not obvious, and the change of the angle between the two vehicles has little effect on it. The change of the lateral safety distance in the vehicle curve is relatively obvious, with an amplitude of about 0.3 m. When the relative speed increases, the longitudinal distance between the two vehicles gradually increases, which is not prone to collision risk, and the lateral safety distance decreases when the vehicle bends. When the included angle between the two vehicles increases, the longitudinal distance between the two vehicles gradually decreases, which renders them prone to collision, and the lateral safety distance increases. The test shows that on the premise of ensuring driving safety, the lateral warning distance can reach 0.94 m and the accuracy of the longitudinal safety warning distance can be increased by 3.8%. Compared with the traditional model, this model predicts the side vehicles based on the relative speed and the angle between the two vehicles so as to improve the reliability of early warning and ensure the driving safety in the state of curve and vehicle lane change.

## 5. Conclusions

In this paper, an intelligent networked car-hailing system based on multi sensor fusion and UWB positioning in complex scenes is proposed. Compared with the traditional car-hailing system, this scheme can achieve high-precision positioning and high lane detection accuracy in complex scenes, and the anti-collision safety warning is more in line with the actual driving characteristics, It provides a guarantee for the safe driving of vehicles in the process of car-hailing, which is of great significance for improving the stability and safety of intelligent car-hailing, and provides a new idea for promoting the development of digital intelligent transportation, which has a good market prospect and application value.

**Author Contributions:** Z.W. (Zhi Wang) conducted experiments and wrote the manuscript; L.Z. contributed to the conception of the study; Y.T. was responsible for testing and put forward constructive amendments; Y.S. summarized the data; Z.W. (Zhenxuan Wu) found some information and summarized the research status at home and abroad. All authors have read and agreed to the published version of the manuscript.

**Funding:** This work was supported by the National Natural Science Foundation of China (Grant Number. 51605215), the China Postdoctoral Science Foundation (Grant Number. 2019T120450), the Qing Lan Project, the Research Foundation of Nanjing Institute of Technology (Grant Number. CKJA201906) and the Science and Technology Innovation Fund Project of Nanjing Institute of Technology (Grant Number. TB202117028)

**Conflicts of Interest:** The authors declare no conflict of interest.

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
