# Peer review of "An Intelligent Networked Car-Hailing System Based on the Multi Sensor Fusion and UWB Positioning Technology under Complex Scenes Condition"

_wevj, doi:10.3390/wevj12030135_

Round 1
Reviewer 1 Report
The paper proposes an intelligent networked car-hailing system based on the multi 3 sensor fusion and UWB positioning technology under complex 4 scenes condition. While the topic is interesting for the community some major modifications are needed before the publication.
1) The authors propose to use UWB technology. Why other technology already proposed for localization and target detection such as FMCW that are typically simpler to implement are not even considered in this work? Also reference related to this topic are missing in the introduction.
2) The references cited by the authors are only 13, too few for a journal paper. A more detailed analysis of the literature is required.
3) Which is the system used to test the UWB technology improvement? More details about the algorithm and the set-up/system used to obtain fig. 5 are required.
3) In Fig.5 what is the real position/reference?
4) In Fig 11-13 legends are missing.
5) To support the improvement of the proposed method a comparison with existing techniques is required. How much do the proposed solutions improve the traditional techniques?
6) Almost no experiments are reported in the paper to support the proposed solution.
7) The quality of figures is too poor for the journal standards.
Author Response
Dear Editors and Reviewers,
Thank you for your letter and for the reviewers' comments concerning our manuscript entitled “An Intelligent Networked Car‐hailing System Based on the Multi Sensor Fusion and UWB Positioning Technology under Complex Scenes Condition”. Those comments are all valuable and very helpful for revising and improving our paper, as well as the important guiding significance to our researches. We have studied comments carefully and have made correction which we hope meet with approval. Revised portion are marked in red in the paper.

Reviewer 2 Report
Corrections are given in the PDF document.

Author Response

(The authors gave the same response as above.)

Round 2
Reviewer 1 Report
While most of the comments have been addressed by the authors, some issues still need to be addressed before publication.
1) According to the answer to the comment 1), the authors state that the antenna beam at MMW is wider than the one at lower frequency, while this parameter mainly depends on the antenna dimensions. In addition, to the reviewer knowledge FMCW radars, if feed with a triangular signal (not the chirp) can estimate both the position and the speed of the target with an architecture that is simpler than the UWB one. Please discuss this.
2) The answer to comment 3) is not complete. No detailed description of the radar system has been added (not even the working frequency).
3) Figure quality has not been improved. I suggest providing vectorial figures or at least increasing the resolution of the ones shown in the paper.
Author Response
Thank you for your letter and for the reviewers' comments concerning our manuscript entitled “An Intelligent Networked Car-hailing System Based on the Multi Sensor Fusion and UWB Positioning Technology under Complex Scenes Condition”. Those comments are all valuable and very helpful for revising and improving our paper, as well as the important guiding significance to our researches. We have studied comments carefully and have made correction which we hope meet with approval. Revised portion are marked in red in the paper. See the attached PDF for the reply to the reviewer's comments.

Reviewer 2 Report
The manuscript has been sufficiently improved to warrant publication in WEVJ.
Author Response
Thank you for your letter and for the reviewers' comments concerning our manuscript entitled “An Intelligent Networked Car-hailing System Based on the Multi Sensor Fusion and UWB Positioning Technology under Complex Scenes Condition”. Those comments are all valuable and very helpful for revising and improving our paper, as well as the important guiding significance to our researches. Finally, thank you for approving our manuscript to be published in WEVJ.